# Mitigation Actions Scenarios Applied to the Dairy Farm Management Systems

**DOI:** 10.3390/foods12091860

**Published:** 2023-04-29

**Authors:** Giulia Rencricca, Federico Froldi, Maurizio Moschini, Marco Trevisan, Lucrezia Lamastra

**Affiliations:** 1Department for Sustainable Food Process (DiSTAS), Università Cattolica del Sacro Cuore, Via Emilia Parmense 84, 29122 Piacenza, Italy; giulia.rencricca@unicatt.it (G.R.); marco.trevisan@unicatt.it (M.T.); 2Department of Animal Science, Food and Nutrition (DiANA), Università Cattolica del Sacro Cuore, Via Emilia Parmense 84, 29122 Piacenza, Italy; maurizio.moschini@unicatt.it

**Keywords:** carbon footprint, life cycle assessment, product environmental footprint, environmental impacts, milk, dairy mitigation

## Abstract

The environmental impacts of the dairy industry, particularly global warming, are heavily influenced by milk production. Thus, there is an urgent need for farm-level actions and opportunities for improvement, implying mitigation strategies. The aim of this paper is to investigate five possible mitigation actions at the dairy farm and which one the farmers were willing to adopt: management and distribution of livestock manure and fertilizers, anaerobic manure treatment, optimization of the herd composition, feed quality, and heat recovery. A life cycle assessment was conducted on 63 farms using the product environmental footprint approach. The latter was divided into four quartiles, from which four representative farms were selected. For each farm, three scenarios have been analyzed considering the reference impact (reference scenario), the application of the mitigation actions (best-case scenario), and what farmers would implement (realistic scenario). Overall, the most effective mitigation actions in the best-case scenario were anaerobic manure treatment and the management and distribution of livestock manure and fertilizers, showing a potential reduction in total environmental impacts of 7–9% and 6–7%, respectively. Farmers’ responses indicated a willingness to implement the latter mitigation strategy better. The optimization of the herd composition, feed quality, and heat recovery reported a range impact reduction between 0.01–5%.

## 1. Introduction

According to the current literature, the dairy sector is responsible for a certain number of impacts [1,2,3], accounting for 4% of the global greenhouse gas (GHG) emissions [4]. Among the complex dairy supply chain, raw milk production is recognized as the most critical contributor to the entire dairy chain’s environmental impact [5], with an overall contribution to the worldwide GHG emissions of 2.7%, including milk production and transportation [4]. Based on 26 studies analyzed by Guzmán-Luna et al., 2022 [6], dairy farms have been identified as the largest contributor to global warming in the dairy sector, accounting for roughly 80% of the total carbon footprint (CF) of dairy products. The primary sources of GHG emissions are constituted by methane (CH_4_) from enteric fermentation and, to a lesser extent, nitrous oxide (N_2_O) from fertilizers used for the on-farm and off-farm feed production [6]. In addition, N_2_O and CH_4_ emissions are released from manure management, and carbon dioxide (CO_2_) is released from the use of inputs at the farm, such as energy and water use [7]. Although CF is a significant environmental parameter, the dairy sector contributes highly to other environmental impacts, considering that it depends on natural resources and other inputs [6]. Among the other relevant environmental impacts, water use, water eutrophication and acidification, photochemical ozone formation, and particulate matter have also been related to the dairy sector [2]. The dairy industry has significant environmental impacts related to cattle production, which are expected to grow with an increment in milk and dairy product demand due to the estimated rise in the global population [7]. Despite this, it has potential for improvement [8,9,10]. Several mitigation strategies at the farm level have been identified as able to decrease the environmental impact, such as optimization of manure management, including manure storage and spreading on field techniques [11,12], as well as its anaerobic digestion through biogas plants [9,13]. To overcome potential impediments to implementing biogas as an energy source [14], collective biogas plants could be implemented as an alternative to the biogas plant existing at each farm. Additionally, increasing feed self-sufficiency can lower the environmental impacts of purchased feed and its transportation [15,16,17]. Similarly, a low feed quality, as the silage chemical composition and the presence of lower digestible forage in the cattle feed, could increase the CH_4_ emissions from enteric fermentation [18,19]. Other mitigation strategies include the implementation of precision farming tools [20,21], energy recovery from milk cooling devices [22], animal husbandry, and milk productivity intensification, which, in particular, involves the number of lactations per cow [23]. In this context, fostering farmers’ awareness of their role in agricultural activities is fundamental. Farmers’ conception of nature influences their environmental acting, but, rather than a more pronounced environmental awareness, their social identity influences farmers’ commitment to pro-environmental actions [24]. Therefore, the aim is to make farmers recognize their role in influencing the environment and activate their sense of conservation responsibility [25]. 

Most of the literature focusing on the dairy sector is dominated by studies in which farmers are involved to collect primary data useful to estimate the environmental impacts of dairy products and different dairy management options [2,9,12,19,26]. In contrast, the present study addresses a specific gap in the literature by examining the willingness to adopt farm-level mitigation strategies, on which limited papers have been focused [27,28]. 

Therefore, the present study aimed to evaluate how different possible mitigation actions can contribute to improving the environmental sustainability of raw milk produced for a specific protected designation of origin (PDO) cheese made in dairy farms in the Po Valley in Northern Italy through a life cycle assessment (LCA) analysis. Northern Italy is representative of national PDO cow cheese making, producing more than 90% of the national PDO cow cheeses [29]. Moreover, Italy has 56 PDO-certified kinds of cheese, accounting for 28% of the total PDO cheese certifications in Europe and positioning the country as one of the leaders in the sector [30]. 

Additionally, this study investigated environmental farmers’ willingness to adopt mitigation actions to reduce the environmental impacts. Due to the restrictions of production rules for PDO products, the feasible modifications in farm activities are limited. 

The present work aimed at achieving the above-explained research objectives by answering the following research questions:What are the most effective mitigation actions that could be applied to reduce the findings of the LCA analysis of milk production destined for the selected PDO cheese?To what extent could the environmental impacts of raw milk production for PDO hard cheese be reduced by adopting the most effective mitigation measures?

## 2. Materials and Methods

### 2.1. Life Cycle Assessment Analysis

The LCA is one of the most commonly used methodologies for assessing the environmental impact of products and processes [31]. A LCA analysis (cradle-to-farm gate) was performed according to the product environmental footprint (PEF) methodology and the product environmental footprint category rules (PEFCR) for dairy products [32,33]. The PEF methodology is based on LCA analysis, aiming to standardize LCA methodology options and establish objective criteria for evaluating product environmental impacts [1]. The PEFCR are product-type-specific rules to perform a PEF-compliant study for a specific sector, which is, in this case, the dairy one [33]. 

Primary data were collected on 63 farms located in the Po Valley in Northern Italy through call surveys and on-farm interviews based on the compilation of a questionnaire. The 63 selected farms were selected to be representative of the production of milk destined for PDO hard cheese making in Northern Italy based on geographic position, the average number of total lactating cows, and milk yield, according to the PEF’s data quality criteria. 

The environmental analysis allowed the creation of average and aggregated datasets and, consequently, the assessment of which impact category contributes to which extent to the environmental footprint of the raw milk production for PDO cheese. The functional unit (FU) was 1 kg/year of fat protein corrected milk (FPCM), and the allocation followed a bio-physical approach between milk and the meat sold [34]. The evaluation of the environmental impact was expressed per FU, according to the following 13 environmental impact indicators: climate change (CC, kg CO_2_-eq), ozone depletion (OD, kg CFC-11eq), ionizing radiation—human health (IR-HH, kBq eq), photochemical ozone formation—human health (POF, kg NMVOCeq), particulate matter formation (PM, disease incidence), acidification (A, mol H + eq), eutrophication freshwater (FE, kg Peq), marine eutrophication (ME, kg Neq), terrestrial eutrophication (TE, mol Neq), ecotoxicity freshwater (FWE, CTUe), land use (LU, pt), water scarcity (WRD, of deprived water), resource use, mineral and metals (M-RD, Kg Sbeq), and fossil resource use (F-RD, MJ). The impact indicators were further normalized and weighted following the Environmental Footprint (EF) method, version 2.0, without tox categories, implemented in the software SimaPro^®^ version 9.0.0.35 (Pré Sustainability, Amersfoort, The Netherlands) [33,35]. 

The collected data were divided into eight main farm processes: water used on the farm, feed purchased, energy, in-farm feed, bedding materials, manure management emissions, enteric fermentation emissions, and barn management emissions. 

### 2.2. Farms Selection and Scenarios Definition

The original dataset of the 63 dairy farms was ordered with respect to the total environmental impacts (weighted results expressed in Pt) and divided into four groups corresponding to four quartiles: Q1 (1st quartile); Q2 (2nd quartile); Q3 (3rd quartile); Q4 (4th quartile). From each quartile, one dairy farm has been selected and considered representative only if its total weighted environmental impact was as close as possible to the median of the quartile to which it belonged. Therefore, each of the four quartiles corresponds to one farm, named Q1, Q2, Q3, and Q4. This selection was made to quantify the potential impact reduction of mitigation actions for different farm management systems. The designated four farms were re-contacted to evaluate possible mitigation actions at the farm level and the willingness of farmers to adopt them. If farmers were not currently using those mitigation techniques, they were asked if they would be interested.

For research purposes, three different scenarios were developed:-The reference scenario, corresponding to the actual environmental impact outcomes of each of the four quartiles farms as a result of the current farmers’ management choices.-The best-case scenario, which considered a maximal possible reduction due to the implementation of different mitigation measures separately at the farm level.-The realistic scenario, which acknowledged the real progress in environmental impact reduction based on the farmer’s possibilities and attitudes to apply the selected mitigation actions based on the questionnaire’s outcomes.

The questionnaire result could be in line with the best-case scenario, representing the farm’s propensity to improve its initial situation through a specific mitigation action, or being in line with the reference scenario, showing a continuance of the farmer’s habitual choices. 

For the best-case scenario and the realistic scenario, the data from the four representative farms were evaluated following the PEF method for 13 impact categories, according to the methodologies explained in Section 2.1. 

### 2.3. Mitigation Action Selection

Based on the original dataset results and the most impacting farm activities, effective mitigation actions applicable by the farms considered were selected from the literature, based on the LCA results through keyword research on Google Scholar, Scopus, and Google research. The publications deemed most suitable to meet the research needs were chosen. Five mitigation action groups were identified to be incorporated in the analysis: management and distribution of livestock manure and fertilizers (Group 1, GR 1), anaerobic manure treatment (biogas) (Group 2, GR 2), optimization of the herd composition (Group 3, GR 3), feed quality (Group 4, GR 4), and heat recovery (Group 5, GR 5). Table 1 reports each mitigation action with the corresponding literature reference(s), the keywords used to find them, the impact category affected by the related emissions, and the percentage of possible environmental impact reduction. The mark (-) in Table 1 indicates whether the farms had already applied the mitigation action, while the asterisks (*) report if the farms would implement the mitigation. 

## 3. Result and Discussion

### 3.1. Life Cycle Inventory Results

The inventory primary collected data are reported in Table 2 as averages for the 63 farms and results for each of the four quartiles allocated to the FU. The 63 farms chosen to represent the Northern Italian PDO production were then divided into four quartiles, from which four representative farms were selected. Table 2 summarizes data on the characteristics of the farms considering the dairy farm management, the on-farm feed and off-farm feed production, and the energy and the bedding materials used on the farm. The four farms represented the possible variability in farm management choices regarding the number of dairy cows, the milk productivity, the quantity of feed produced in-farm, and the feed purchased. 

### 3.2. Original Dataset and four Quartiles of Environmental Impacts 

Table 3 reports the characterized results of the environmental impact assessment for milk production related to the 63 original dataset farms and the four quartiles of farms obtained through applying the PEF methodology and PEFCR for dairy products rules. 

The four quartiles showed varying degrees of environmental effect due to differences in farm characteristics, as shown in Section 3.1. 

Figure 1 displays the considered farm process contribution to the most affected impact categories for the 63 original dataset farms. The results showed that the most major overall environmental impacts resulted from feed purchased (34%), followed by in-farm feed production (25%) and emissions from manure management (16%), enteric fermentation (12%), and barn management (6%), in line with Froldi et al., 2022 [2]. Energy, bedding materials, and water used on farms counted, together, for 7% of the total impact. The most affected impact categories were CC (32%), WRD (25%), TE (11%), LU (7%), PM (6%), and POF (6%) (Figure 1), similar in part to what found by Lovarelli et al., 2022 [9]. The A, ME, and M-RD impacts accounted for 3–4% of the overall impact. At the same time, OD, IR-HH, FE, and F-RD resulted in less than 1%, so they are negligible in this context, not shown in Figure 1. The CC was affected for 37%, 31%, and 23% by enteric fermentation, feed purchased, and manure management, respectively. The WRD contribution was mainly influenced by in-farm feed production, covering 73% of the impact. In comparison, the percentage contribution decreases to 22% when looking at feed purchased and 4% for water used on the farm. The potential impact of TE was impacted by manure management (70%) and feed purchased (20%). The LU impact was highly influenced by feed purchased (63%) and in-farm feed (34%). Emissions from barn management were the primary factor affecting POF (75%). Bedding materials emissions were negligible and are not reported in Figure 1.

As for the 63 original dataset outcomes, the emissions from feed purchased, feed produced on-farm, and manure management were the highest also for the four farm quartiles (Appendix A). 

Bedding materials impacts are not displayed since their emissions are negligible (<1%).

### 3.3. Mitigation Actions Results for the Four Quartiles Farms

The four farms in this study were designated with Q1, Q2, Q3, and Q4, representing the first, second, third, and fourth quartiles, respectively. For each farm, three scenarios are presented: the reference scenario, the best-case Scenarios applied to each of the five groups of mitigation measures, and the realistic scenario. Figure 2 reports the results per each representative farm for the different scenarios as a single total score expressed in Pt. Following the farmers’ responses, the reduction in emissions and, consequently, in environmental impact due to adopting mitigation actions, were quantified by forming the realistic scenario. Appendix A reports the numeric weighted results presented in Figure 2. Table 4, Table 5, Table 6, Table 7 and Table 8 present in percentage for each mitigation group, the effect of the best-case scenario on each impact category, and the total impact compared to the reference scenario results. Table 4 also displays the results for the realistic scenario. 

The disparity in the possible reduction among the four quartiles is attributed to the reference farms’ different management systems. For quartile Q1, the mitigation strategies GR 3 and GR 5 were not applicable and considered to be already optimal for this farm. 

Overall, as shown in Table 4, Table 5, Table 6 and Table 7, manure management emissions had the greatest potential for a decrease in this context. At the same time, the mainly affected impact was CC for all three scenarios, except for the best-case scenario GR 2, as noticeable from Figure 2. 

The application of the mitigation GR 1 decreased for the four quartiles the impacts of manure management emissions, barn management emissions, feed purchased, except for Q4, and in-farm feed only for Q3 and Q4, diminishing all the environmental impacts considered by 6–7% in total (Figure 2). Through the introduction of GR 1, manure management emissions process showed a similar potential reduction in all the four quartiles, ranging between 24–30%, a lesser impact compared to the reference scenario. These results are also confirmed by Sajeev et al., 2018 [42], who reported reductions in GHG and ammonia (NH_3_) emissions along the entire manure management chain. In our conditions, the mitigation strategy lowered the potential impact of A and TE between 46–54% (Table 4) as a consequence of the NH_3_ reduction due to the best agriculture practices for manure spreading [36] and manure storage [43], nitrogen fertilizers spreading, and use of slow-release fertilizers for urea [39]. Similarly, the emissions from barn management could be decreased by 17–19%, reducing the potential impact of POF, referred to the emission of particulate matter (<2.5 µm) and non-methane volatile organic compounds (NMVOC) [2]. This mitigation action reduced market dependence on feed purchased and raw materials due to a hypothetical increase in self-produced crop yields. This reduction of 2–8% was not observed for the Q4 farm, for which raw material and feed purchase quantities remained unchanged, as did the resulting emissions. The application of GR 1 for the purchase of feed reduced all impact categories considered in this context by 5–29% (Table 4), as manure management practices (storage, spreading of manure and fertilizer on the soil) reduce NH_3_ emissions, as presented above, leading to a potential increase in crop yields due to increased nitrogen availability to the field. In this regard, Sefeedpari et al., 2019 [44], in a technical, environmental, and economic study on manure management, state that good agricultural practices of this resource condition are related to nitrogen availability of crops. Given the high impact associated with the purchase of feed, particularly from geographic areas affected by land use change [19], GR 1 actions and proper management of the farm’s raw materials make it possible to reduce the share of feed purchased on the market by enhancing self-production. As Gaudino et al., 2018 [45] pointed out, this approach represents a good opportunity to increase feed protein self-sufficiency. However, the increase in the in-farm feed also led to an increase in emissions from crop production for those farms, particularly Q1 and Q2, that did not have high yields. In fact, for Q1 and Q2, the impact of WRDs increased by 17–8%, respectively (Table 4), as higher amounts of water for irrigation were justified to achieve higher yields in proportion to the lower amount of feed purchased. 

Farmers would be more eager to apply GR 1 as mitigation action because it has a high potential for environmental impact reduction, even if it involves investments to buy storage covers and equipment to be used as injectors for fertilizer spreading.

On-farm anaerobic digestion of manure effectively reduces greenhouse gas emissions and non-renewable energy consumption [46]. In this context, indeed, the mitigation strategy of GR 2 led to the highest environmental impact reduction for all the quartiles, as mentioned above, with a decrease of 7–9% (Figure 2). These results are distinguished in the macro area of manure management emissions by reducing the total impact by 44–50%, and consequently the effects on POF and CC indicators for this process are highly reduced by 92–96% (Table 5). CH_4_ emissions from conventional manure storage were reduced due to the creation of bioenergy and digestate, which can be used as fertilizer for agricultural land. The high reduction potential is similar to the findings of Lovarelli et al., 2022 [9], for whom anaerobic manure treatment can cause a reduction of GHG emissions by 30% per FU. Indeed, this mitigation strategy is widely known as an effective measure to reduce farm impacts [9,40]. Pexas et al., 2020 [47] stated that there are no one-size-fits-all solutions to improve environmental and economic performance in the livestock sector, but mitigation strategies must be planned and considered. In this regard, the anaerobic digestion systems require large initial capital investments for constructing the biogas plant, along with maintenance costs. However, in the present scenario, potential environmental benefits have been calculated, assuming that the manure would be sent to a biogas plant not owned by the farm itself, but to a biogas consortium plant. However, it requires structural investments and natural resources [48] to transport the slurry to a collective anaerobic digestion plant and the digestate to the farm of origin, also considering the distance between production sites [27]. 

The introduction of the mitigation strategy GR 3, which was not calculated for Q1 because the herd composition was estimated to be already optimal, reduced the total impact of Q2, Q3, and Q4 quartiles by 2–5%, lowering to a similar extent all the impact categories considered (Figure 2). In accordance with Knapp et al., 2014 [49], the heat stress abatement, disease control and treatment, performance-enhancing technologies, and management solutions toward the improvement of animal reproductive performance are estimated to lower environmental impact per kg of FPCM, depending on the cow’s genetic potential. Additionally, the increase in milking frequency improves the environmental performance of the herd when referred to the CF of milk [50], as well as the increase in milk yield [51]. In contrast, a reduction of the age at first calving and of the replacement rate, as for the improvement in fertility, can reduce CH_4_ emissions by up to 24% [52]. Özkan Gülzari et al., 2018 [53], stated that compromised animal health status is responsible for losses in both productivity and profitability on dairy farm activity. For these reasons, improved udder health and milk quality would lead to a reduction of the GHG emission intensity of the herd. In the context of the mitigation actions referred to GR 3, optimizing the composition of the herd implied maintaining the number of lactating cows constant while minimizing the number of non-productive animals, increasing farm management efficiency. Indeed, the management of cow breeding can be seen as a mitigation intervention, acting at different levels that influenced almost all the studied processes, as the number of animals has changed, with, however, the highest reduction for bedding materials (2–13%), feed purchased (2–7%), and enteric fermentation emissions (5–6%) (Table 6). The implementation of this mitigation only concerns the better management of the reared herd, not involving structural investments, and, hence, it is easy to be implemented. However, the environmental benefit would be observed only over the long term.

The GR 4 investigated how the quality of feeds used in diet formulation affected the estimated digestible energy and CH_4_ output, mainly from enteric fermentation, but also from manure. The main factors determining enteric CH_4_ emissions are feed intake, forage digestibility and quality, and dry matter intake level [18,41]. According to Tullo et al., 2019 [21], continuous monitoring of feed quality parameters can reduce CF of milk by reformulating diets over time. In this specific scenario, the farms considered already had a good quality of in-farm feeds; therefore, the application of this mitigation action showed a reduction of the total average impact in a range between 1 and 3% for the four quartiles considered (Figure 2). The process reduction addressed by the mitigation measure was emissions from enteric fermentation in the range of 3 to 11% and emissions from manure management in the range of 7 to 10%, mainly reducing POF and CC potential impacts (Table 7). The findings agree with Caro et al., 2016 [54], who found a potential reduction of enteric emissions in Europe of 10% by modifying the cattle diet, using high-quality forages, and consequently reducing the fibre content in the dairy cows’ diets [55]. 

Feed quality, therefore, influences both enteric and manure CH_4_ production. It is possible to state that the higher the feed quality, the lower the CH_4_ emissions from enteric fermentation due to enhancing the feed conversion ratio and, consequently, less indigestible feed in the manure [10,19]. These results align with a study by Knapp et al., 2014 [49], which examined a significant potential for reducing CH_4_ emissions through high-quality feeds and, thus, good performance in reducing the CC indicator.

This mitigating approach, however, has limitations and necessitates significant investments, such as external technical assistance, analysis expenses, and the use of analytical feed composition in diet formulation.

The GR 5 mitigation measure resulted in a reduction in the environmental impacts of only 0.01–0.04%, reducing the emissions derived from energy, mainly from M-RD impact (Figure 2). Few studies have been conducted on this in the scientific literature. Schader et al., 2014 [22], examined a GHG reduction potential at the dairy farm level of 0.14% due to heat recovery from the milk tank. In this respect, the mitigation consisted of heat recovery from the milk cooling process, which, as an example, could be used to heat water to wash the milking equipment [56]. It has a straightforward implementation with low economic investment, but a low reduction potential compared to the other mitigation groups. In fact, only the amount of fossil energy used in heating the water for washing machinery is replaced by a milk tank equipped with a heat exchanger.

However, the use of fossil energy in this context was very modest, and its replacement would have a limited effect on the reduction of the total impact of milk, as confirmed in Table 8. 

### 3.4. The Realistic Scenario Results

As shown in Table 1, the farm representatives of the four quartiles are willing to employ different mitigation actions marked with the asterisk. Nevertheless, Mitigation GR 1 found greater interest by farmers for the possible ease of implementation. Therefore, in the realistic scenario, each of the four farms would employ only the GR 1 to varying degrees (Table 1), producing different results (Table 4). 

For the Q1 farm, the realistic scenario showed overall 1% higher results than the best-case scenario (Figure 1) for GR 1 due to 13% higher emissions for barn management emissions, resulting in higher POF emissions (Table 4). Indeed, the farm preferred to continue with the conventional manure application approach and fertilizers incorporation, in contrast to the best-case scenario, which considered slurry injection (closed-slot) and closed-slot injection of fertilizers. In their reference scenario, Q1 favored the formation of natural crust during manure storage. 

Q2 farm would partly apply the GR 1 mitigation strategy, so the results of the realistic scenario correspond to those obtained from the best-case scenario (GR 1), except for feed purchased and in-farm feed. The in-farm feed category was characterized by +6% and the feed purchased process by −4% environmental impact in the best-case scenario compared to the realistic scenario (Table 4). Nevertheless, the total impact reduction as an application of the mitigation measure stands a 6% in the realistic scenario, as well as in the best-case scenario (GR 1) (Figure 1). 

For farm Q3, farmers were not willing to entirely apply the GR 1, preferring the conventional manure application and no improvement in manure storage. This farm would implement the incorporation of fertilizers, which, however, decreases in lesser extent the NH_3_ emissions compared to the closed-slot injection practice, as shown in Table 1. Therefore, the realistic scenario resulted higher than the application of the corresponding best-case scenario, which would decrease by 7% the total impact of the farm by diminishing mostly TE impact (Figure 2).

The Q4 would partly employ the mitigation GR 1, as well, resulting in a realistic scenario 3% higher compared to the best-case scenario total impact (Figure 2). In particular, the complete application of the mitigation measure would result in a significant reduction in the impacts of the TE, ME, POF, and A indicators (Table 4). In fact, the emissions from manure management and barn management were higher in the realistic scenario. These results are probably due to the farm management’s choice of favoring the natural crust of manure storage and the open-slot as the application of manure, with less reduction in NH_3_ emissions than the closed-slot (Table 1). Similarly, Q4 would apply fertilizer incorporation as fertilizer application instead of injection, which would be more efficient.

Abbreviations: Reference: reference scenario; BS GR 1: best-case scenario applying group 1; BS GR 2: best-case scenario applying group 2; BS GR 3: best-case scenario applying group 3; BS GR 4: best-case scenario applying group 4; BS GR 5: best-case scenario applying group 5; Realistic: realistic scenario.

## 4. Conclusions

The environmental impact of milk produced for PDO hard cheese-making was assessed. Four dairy farms were chosen corresponding to four quartiles with respect to the total environmental impact of 63 farms in Northern Italy. Five mitigation measures have been selected, differing in mitigation potential and feasibility. Three different scenarios have been analyzed for each quartile considering the reference farm impact (reference scenario), the application of the five mitigation actions (best-case scenario), and what farmers would implement as mitigation (realistic scenario). 

Overall, climate change was the impact indicator most affected by farm activities, approximately 32% of the total impact, mainly by enteric fermentation emissions, feed purchased, and manure management emissions. The latter showed the greatest improvement potential.

Anaerobic manure treatment was the most effective mitigation action in terms of total environmental impact reduction (7–9%), with relatively easy implementation, reducing half the manure management emissions and its effect on climate change. However, farmers’ responses indicated an intention to implement better management and distribution of manure and fertilizers in their farm management as a mitigation strategy. This mitigation has higher feasibility, with a considerable potential reduction as well (6–7%), being particularly effective in reducing manure management emissions. Additional environmental benefits from this mitigation came from improved barn management and feed purchased-related emissions. Despite the optimization of herd composition, this would not require major efforts for the implementation, and the environmental benefits that can be obtained were rather limited (2–5%). Increasing the feed quality and the heat recovery are considered difficult to implement, with scarce environmental impact reduction (1–3% and 0.01–0.02%, respectively). 

In general, increasing farmers’ awareness of the implications of their management decisions is necessary. Nevertheless, the development of mitigation actions is challenging. Site-specific conditions are variable, and despite the presence of PDO regulations, mitigation measures cannot be standardized for every farm. Future research should be focused on studying the possible reduction of the environmental impact by the interaction of several mitigation actions at the dairy farm level. In this case, it would be appropriate to study not only the environmental benefits, but also the economic sustainability linked to substantial investments for purchasing equipment or structural adaptation. 

## Figures and Tables

**Figure 1 foods-12-01860-f001:**
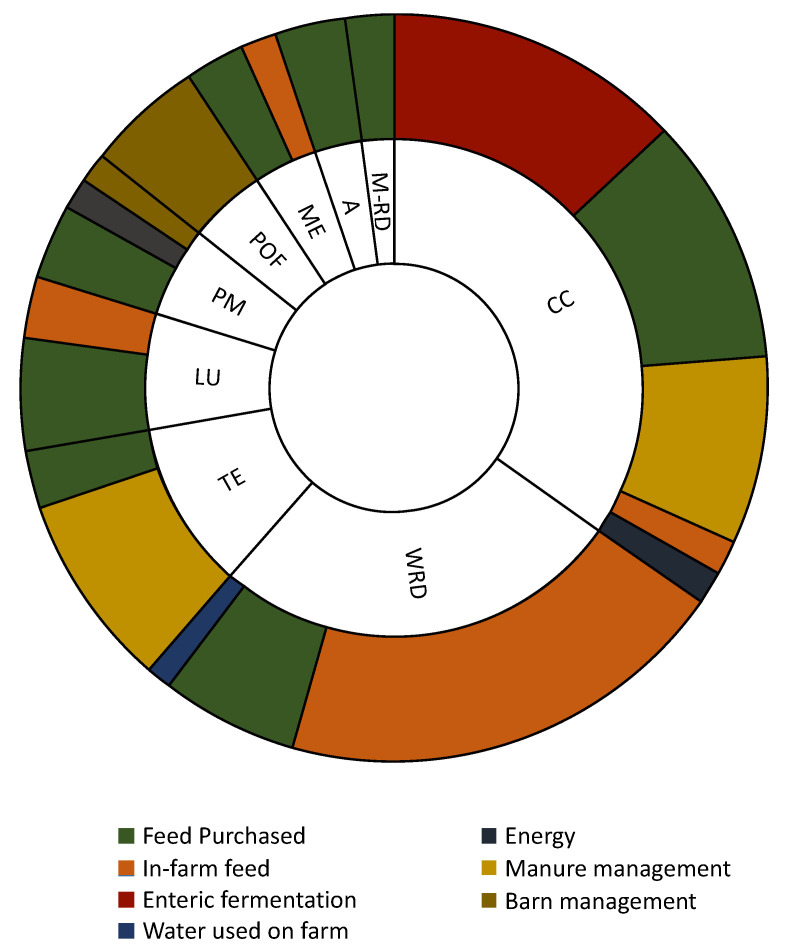
Environmental contribution of the different processes considered to produce 1 kg FPCM milk for the original dataset (63 dairy farms). Abbreviations: CC (Climate change); POF (Photochemical ozone formation, human health); PM (Particulate matter formation); A (Acidification, terrestrial and freshwater); ME (Eutrophication, marine); TE (Eutrophication, terrestrial); LU (Land use); WRD (Water scarcity); M-RD (Resource use, mineral and metals).

**Figure 2 foods-12-01860-f002:**
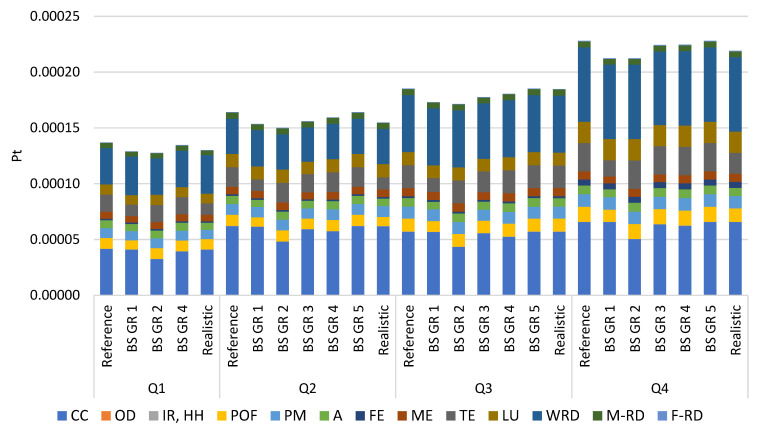
Potential reduction of impacts by applying the five mitigation actions for the different Scenarios in each quartile Q1, Q2, Q3, and Q4.

**Table 1 foods-12-01860-t001:** Groups of mitigation actions considered and potential percentage of reduction of emitted substances based on a detailed study review research.

Mitigation Group	Mitigation Action	Emissions	Reduction (%)	^1^ Q1	^1^ Q2	^1^ Q3	^1^ Q4	Literature	Literature Keywords
Management and distribution of livestock manure and fertilizers (GR 1)	Natural crust formation	NH_3_	40	-			*		
	Straw to encourage natural crust formation		40						
	Tight/rigid lids		>80, 80	*	*				
	Flexible cover		80–90						
	Alternative baghouse to the concrete tank		80						
	Plastic sheets		>60, 60						
	Floating covers		Up to 40						
	Band spreading slurry (trailing hose)		30–35					[36,37]	NH_3_ emission; best available techniques; manure application; fertilizer application; low-emission manure.
	Band spreading (trailing shoe)		30–60						
	Injection slurry (open slot)		70				*		
	Injection slurry (closed slot)		80–90		*				
	Incorporation of surface-applied slurry		30–90						
	Use of urease and nitrification inhibitors		40–70						
	Slow-release fertilizer		ca.30	*	*				
	Closed-slot injection		80–90						
	Incorporation		50–80	*	*	*	*		
	Fertigation		40–70						
	Substitution with ammonium nitrate		up to 90						
	Reduction of mineral fertilizer usage (15%)	N_2_O	5					[38,39]	Reduction in synthetic fertilizer; reduction in N_2_O emissions; mitigation strategy to reduce nitrate emission; mitigation strategies at farm level.
Anaerobic manure treatment (GR 2)	Biogas plant at the farm or collective biogas plant	CH_4_	23–36 of total GHG	*	*	*		[40]	Anaerobic digestion; biogas; manure valorization; GHG reduction with anaerobic digestion treatment.
Optimization of the herd composition (GR 3)	Improve fertility	GHG	>20	*	*	*	*	[10]	Fertility and greenhouse gas emission; herd management; environmental impact of milk production and fertility.
	Increase lactations per cow		4.5 (increase one lactation/cow)					[23]	Environmental impact of milk production, LCA dairy farm, number of lactations and environmental impact.
Feed quality (GR 4)	Forage quality parameters	CH_4_	11	*	*	*	*	[18,41]	Forage quality; methane emission, enteric methane; feed intake quality.
Energy recovery (GR 5)	Milk-tank heat exchanger	GHG	0.14	*	*		*	[22]	Energy use on farm, energy source, energy consumption at farm level

^1^ Farms ranked into four quartiles: Q1 (1st quartile); Q2 (2nd quartile); Q3 (3rd quartile); Q4 (4th quartile). * The farm would implement the mitigation action. - The farm already implements the mitigation action. Abbreviations: NH_3_: ammonia; N_2_O: nitrous oxide; CH_4_: methane; GR 1: group 1; GR 2: group 2; GR 3: group 3; GR 4: group 4; GR 5: group 5.

**Table 2 foods-12-01860-t002:** Inventory data of the original dataset and the four quartiles.

		Original Dataset	^1^ Farms			
		63 Farms	Q1	Q2	Q3	Q4
Characteristics	Units	Mean	Value			
Dairy farm management						
Dairy cows	Cows farm ^−1^ year ^−1^	135	217	235	45	120
^a^ Other cattle	Cows farm ^−1^ year ^−1^	153	195	295	47	133
Average milk production	t FPCM farm ^−1^ year ^−1^	1474.8	3063.2	2928.4	443.4	951.4
Average meat production	t meat farm ^−1^ year ^−1^	29.4	48.3	70.1	7.6	11.9
Milk fat content	%	3.9	4.1	4.0	4.2	4.3
Milk protein content	%	3.4	3.6	3.3	3.8	3.7
Allocation						
Raw milk	%	87.0	90.0	86.0	89.0	92.0
Meat co-product	%	13.0	10.0	14.0	11.0	8.0
Water used on the farm	m^3^ farm^−1^ year^−1^	8575.2	14,882.1	15,068.4	2094.1	7562.1
In-farm feed						
Usable agricultural area	ha	55.3	50.7	38.1	34.4	87.7
Alfalfa	ha	7.1	13	3.7	2.7	-
Forages	ha	22.3	8.0	18.9	15.4	48.3
Cereals silages	ha	4.1	4.7	15.5	-	16.4
Cereals grain	ha	0.8	-	-	-	-
^b^ Corn	ha	28.3	25	34.4	15.4	38.1
^b^ Other silages	ha	4.3	4.7	-	-	-
Nitrogen fertilizers	kg farm^−1^ year^-^	4509.5	1035.0	5393.2	3796.0	1423.5
Potassium fertilizers	kg farm^−1^ year^-^	192.6	-	-	1520.0	1423.5
Phosphate fertilizers	kg farm^−1^ year^-^	272.6	-	-	2160.0	1423.5
Pesticides	kg farm^−1^ year^-^	185.28	334.6	226.6	101.7	250.2
Irrigation water	m^3^ farm^−1^ year^-^	146,641.2	175,467.8	201,335.5	77,598.7	155,980.7
Off-farm feed						
Alfalfa	kg farm^−1^ year^-^	53,256.2	371,250.0	100,000.0	15,000.0	-
^c^ Forages	kg farm^−1^ year^-^	110,355.4	150,000.0	409,500.0	-	156,000.0
^d^ Corn	kg farm^−1^ year^-^	146,028.2	402,010.0	80,510.0	30,000.0	175,200.0
Corn silage	kg farm^−1^ year^-^	225,697.7	1,237,500.0	600,000.0	-	-
Wholemeal corn mash	kg farm^−1^ year^-^	23,294.6	-	-	-	-
^e^ Other silages	kg farm^−1^ year^-^	68,026.2	-	-	-	-
^f^ Protein feeds	kg farm^−1^ year^-^	96,675.4	-	99,050.0	41,063.0	109,500.0
Compound feed	kg farm^−1^ year^-^	253,568.3	252,900.0	1,206,462.5	51,850.0	110,950.0
Mineral and vitamins	kg farm^−1^ year^-^	14,982.3	17,950.0	5725.0	750.0	-
Energy						
Electricity	kWh farm^−1^ year^-^	64,019.9	117,736.0	162,000.0	15,974.0	21,516.0
Diesel	lt farm^−1^ year^−1^	30,131.4	44,414.8	47,324.3	11,789.4	19,041.0
Bedding materials						
Cereal straw	kg farm^−1^ year^−1^	63,670.8	120,000.0	12,780.0	90,000.0	10,000.0
Sawdust and woodchips	kg farm^−1^ year^−1^	2712.2	-	-	-	50,000.0

^1^ Farms ranked into four quartiles: Q1 (1st quartile); Q2 (2nd quartile); Q3 (3rd quartile); Q4 (4th quartile). ^a^ The category “other cattle” referred to dry cows, heifers, young heifers (from weaning to 12 months of age), and calves (from birth until weaning). ^b^ The “corn” and “corn silage” are considered the first and second harvesting crops. ^c^ The “forages” referred to polyphyte hay, lolium multiflorum, and cereals straw feeds. ^d^ The “corn” referred to the corn flour and corn flakes feed. ^e^ The “other silages” referred to sorghum and wheat silage feeds. ^f^ The “protein feeds” referred to soybean meal, soy flakes, linseed hulls, and dehulled sunflowers.

**Table 3 foods-12-01860-t003:** Characterized results expressed per 1 kg of FPCM of the original dataset (63 dairy farms) and the four quartiles.

Impact Indicator	Units	Original Dataset	^1^ Farms			
		63 Farms	Q1	Q2	Q3	Q4
Climate change (CC)	kg CO_2_-eq.	1.91 × 10^0^	1.45 × 10^0^	2.17 × 10^0^	1.69 × 10^0^	2.30 × 10^0^
Ozone depletion potential (OD)	kg CFC-11-eq.	6.99 × 10^−10^	1.04 × 10^−9^	6.99 × 10^−10^	7.93 × 10^−10^	8.04 × 10^−10^
Ionizing radiation, human health (IR-HH)	kBq U^235^-eq.	1.48 × 10^−2^	1.41 × 10^−2^	1.67 × 10^−2^	1.22 × 10^−2^	1.41 × 10^−2^
Photochemical ozone formation, human health (POF)	kg NMVOC-eq.	8.35 × 10^−3^	7.71 × 10^−3^	7.96 × 10^−3^	8.06 × 10^−3^	1.08 × 10^−2^
Particulate matter formation (PM)	disease inc.	6.45 × 10^−8^	5.88 × 10^−8^	6.44 × 10^−8^	5.84 × 10^−8^	7.41 × 10^−8^
Acidification, terrestrial and freshwater (A)	mol H^+^-eq.	5.92 × 10^−3^	5.93 × 10^−3^	6.04 × 10^−3^	5.41 × 10^−3^	6.61 × 10^−3^
Eutrophication, freshwater (FE)	kg P-eq.	1.50 × 10^−4^	1.19 × 10^−4^	1.26 × 10^−4^	3.37 × 10^−4^	4.62 × 10^−4^
Eutrophication, marine (ME)	kg N-eq.	6.41 × 10^−3^	5.70 × 10^−3^	6.13 × 10^−3^	7.55 × 10^−3^	6.60 × 10^−3^
Eutrophication, terrestrial (TE)	mol N-eq.	8.56 × 10^−2^	6.93 × 10^−2^	7.97 × 10^−2^	8.19 × 10^−2^	1.15 × 10^−1^
Land use (LU)	Pt	1.94 × 10^2^	1.43 × 10^2^	1.87 × 10^2^	2.37 × 10^2^	3.03 × 10^2^
Water scarcity (WRD)	m^3^ depriv.	5.36 × 10^0^	4.15 × 10^0^	4.02 × 10^0^	7.99 × 10^0^	8.47 × 10^0^
Resource use, mineral and metals (M-RD)	kg Sb-eq.	2.86 × 10^−7^	2.50 × 10^−7^	2.96 × 10^−7^	4.98 × 10^−7^	4.51 × 10^−7^
Resource use, fossils (F-RD)	MJ	3.76 × 10^0^	3.37 × 10^0^	3.95 × 10^0^	3.78 × 10^0^	3.74 × 10^0^

^1^ Farms ranked into four quartiles: Q1 (1st quartile); Q2 (2nd quartile); Q3 (3rd quartile); Q4 (4th quartile).

**Table 4 foods-12-01860-t004:** Best-case scenario (BS) and realistic scenario (RS) for each quartile when applying mitigation group 1—“Management and distribution of livestock manure and fertilizers”.

					Group 1										
Quartiles					* Manure Management Emissions (%)							Total (%)			
		CC	OD	IR, HH	POF	PM	A	FE	ME	TE	LU	WRD	M-RD	F-RD	
Q1	BS						−46			−46					−24
	RS						−46			−46					−24
Q2	BS						−53			−53					−24
	RS						−53			−53					−24
Q3	BS						−51			−51					−26
	RS														
Q4	BS						−54			−54					−30
	RS						−33			−33					−19
* Barn management emissions (%)															
Q1	BS				−20										−17
	RS				−5										−4
Q2	BS				−23										−19
	RS				−23										−19
Q3	BS				−23										−18
	RS														
Q4	BS				−24										−19
	RS				−15										−11
* Feed purchased (%)															
Q1	BS	−5	−9	−9	−8	−8	−8	−6	−8	−8	−7	−10	−8	−6	−8
	RS	−5	−9	−9	−8	−8	−8	−6	−8	−8	−7	−10	−8	−6	−8
Q2	BS	−2	−2	−5	−5	−6	−6	−3	−4	−6	−3	−9	−4	−8	−4
	RS														
Q3	BS	−1	−6	−1	−2	−3	−4	−1		−4	−5		−1	−29	−2
	RS														
Q4	BS		−1								−1			−3	
	RS														
* In-farm feed (%)															
Q1	BS	−2		−5	−7	−10	−34		−2	−81		17	−11	−10	12
	RS	−2		−5	−7	−10	−27		−2	−64		17	−11	−10	12
Q2	BS	−8		−11	−11	−13	−41		−4	−83		8	−15	−12	4
	RS	−8		−11	−11	−13	−34		−4	−65			−15	−12	−2
Q3	BS	−6		−12	−14	−16	−49		−6	−85			−17	−16	−2
	RS						−29			−59					−1
Q4	BS						−4		−37	−41					−2
	RS						−3			−29					

* The percentage value referring to the impact indicator is not present when the reference value is zero or the impact indicator is not changed by the mitigation measure.

**Table 5 foods-12-01860-t005:** Best-case scenario for each quartile applying mitigation group 2—“Anaerobic manure treatment”.

				Group 2										
Quartiles				* Manure Management Emissions (%)							Total (%)			
	CC	OD	IR, HH	POF	PM	A	FE	ME	TE	LU	WRD	M-RD	F-RD	
Q1	−92			−95										−44
Q2	−94			−96										−50
Q3	−93			−96										−45
Q4	−92			−95										−40

* The percentage value referring to the impact indicator is not present when the reference value is zero or the impact indicator is not changed by the mitigation measure.

**Table 6 foods-12-01860-t006:** Best case scenario for each quartile applying mitigation group 3—“Optimization of herd composition”.

				Group 3										
Quartiles				* Manure Management Emissions (%)							Total (%)			
	CC	OD	IR, HH	POF	PM	A	FE	ME	TE	LU	WRD	M-RD	F-RD	
Q2	−3					−8			−8					−5
Q3	2					−10			−10					−4
Q4	−3													−1
* Barn management emissions (%)														
Q2				−6	−6									−6
Q3				−6	−8									−6
Q3				0	−3									−1
* Feed purchased (%)														
Q2	−33	−28	−11	−23	−14	−11	−19	−8	−10	−30	−23	−17	−14	−7
Q3	−4	−8	−8	−7	−9	−9	−6	−9	−9	−6	−10	−8	−10	−7
Q4	−1	14	1	−2	−1		−4	−5	0	−3	−7	−1	7	−2
* Enteric fermentation emissions (%)														
Q2	−6			−6										−6
Q3	−4			−4										−4
Q4	−5			−5										−5
* Water used (%)														
Q2	−3	−3	−3	−3	−3	−3	−3	−3	−3	−3	−3	−3	−3	−3
Q3	−4	−4	−4	−4	−4	−4	−4	−4	−4	−4	−4	−4	−4	−4
Q4	−2	−2	−2	−2	−2	−2	−2	−2	−2	−2	−2	−2	−2	−2
* Bedding materials (%)														
Q2	−6	−8	−7	−7	−8	−8	−8	−8	−8	−8	−8	−7	−8	−8
Q3	−12	−13	−13	−13	−13	−13	−13	−13	−13	−13	−13	−12	−13	−13
Q4	−1	−2	−1	−1	−2	−2	−2	−2	−2	−2	−2	−1	−1	−2

* The percentage value referring to the impact indicator is not present when the reference value is zero or the impact indicator is not changed by the mitigation measure.

**Table 7 foods-12-01860-t007:** Best-case scenario for each quartile applying mitigation group 4—“Feed quality”.

				Group 4										
Quartiles				* Manure Management Emissions (%)							Total (%)			
	CC	OD	IR, HH	POF	PM	A	FE	ME	TE	LU	WRD	M-RD	F-RD	
Q1	−18			−21										−9
Q2	−18			−20										−10
Q3	−17			−19										−8
Q4	−15			−18										−7
* Enteric fermentation emissions (%)														
Q1	−3			−3										−3
Q2	−10			−10										−10
Q3	−11			−11										−11
Q4	−3			−3										−3

* The percentage value referring to the impact indicator is not present when the reference value is zero or the impact indicator is not changed by the mitigation measure.

**Table 8 foods-12-01860-t008:** Best-case scenario for each quartile applying mitigation group 5—“Heat recovery”.

				Group 5										
Quartiles				* Energy (%)							Total (%)			
	CC	OD	IR, HH	POF	PM	A	FE	ME	TE	LU	WRD	M-RD	F-RD	
Q2	−1					−1						−4		−1
Q3		−1	−1									−2		

* The percentage value referring to the impact indicator is not present when the reference value is zero or the impact indicator is not changed by the mitigation measure.

## Data Availability

Data is contained within the article or Appendix A.

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
