# Peer review of "Mitigation Actions Scenarios Applied to the Dairy Farm Management Systems"

_foods, 2023, doi:10.3390/foods12091860_

Round 1

Reviewer 1 Report

The study on the mitigation actions for reducing the environmental impacts of the dairy industry is a timely and important contribution to the field. The use of primary data from 63 farms and the Life Cycle Assessment using the Product Environmental Footprint approach is commendable.

The selection of four representative farms from the original dataset and the analysis of three scenarios for each farm provides a comprehensive picture of the potential mitigation actions.

The findings of the study show that anaerobic manure treatment and the management and distribution of livestock manure and fertilizers are the most effective mitigation actions, with a potential reduction in total environmental impacts of 7-9% and 6-7%, respectively. The farmers' responses indicated a willingness to implement the latter mitigation strategy better.

The study highlights the importance of farm-level actions and opportunities for improvement to address the environmental impacts of milk production. However, there are some limitations to the study that need to be addressed. The sample size of 63 farms may not be representative of the entire dairy industry, and the selection of only four representative farms may not accurately reflect the diversity of dairy farming practices. Additionally, the study only considers five mitigation actions, and there may be other potential actions that could be effective in reducing environmental impacts. Despite these limitations, the study provides valuable insights into the potential mitigation actions for reducing the environmental impacts of the dairy industry.

The findings of the study could be useful for policymakers, dairy farmers, and other stakeholders in developing strategies to improve the sustainability of the dairy industry.

Overall, this is a well-conducted study that addresses an important environmental issue and provides practical solutions for reducing the environmental impacts of milk production.

its just minor grammer issues 

Author Response

We thank you for your suggestions for improvements to manuscript foods-2362042. The manuscript has been edited in several parts, and grammar problems have been resolved.

The study on the mitigation actions for reducing the environmental impacts of the dairy industry is a timely and important contribution to the field. The use of primary data from 63 farms and the Life Cycle Assessment using the Product Environmental Footprint approach is commendable.

The selection of four representative farms from the original dataset and the analysis of three scenarios for each farm provides a comprehensive picture of the potential mitigation actions.

The findings of the study show that anaerobic manure treatment and the management and distribution of livestock manure and fertilizers are the most effective mitigation actions, with a potential reduction in total environmental impacts of 7-9% and 6-7%, respectively. The farmers' responses indicated a willingness to implement the latter mitigation strategy better.

The study highlights the importance of farm-level actions and opportunities for improvement to address the environmental impacts of milk production. However, there are some limitations to the study that need to be addressed. The sample size of 63 farms may not be representative of the entire dairy industry, and the selection of only four representative farms may not accurately reflect the diversity of dairy farming practices.

The 63 farms reported in the manuscript refer to a representative sample of dairy farms in Northern Italy. This sample was defined within the framework of the European LIFE TTGG project (The Thought Get Going, 2017-2022, LIFE 16 ENV/IT/000225). Therefore, the manuscript focuses on representative milk production in the Po Valley in a specific case study. As added in the manuscript's revision, Northern Italy is characterized by producing more than 90 percent of PDOs derived from cow's milk processing nationally. Please see the lines 84-87.

Another specific research could support an extension of data collection in the stratification of farms at the national level to study the propensity to adopt mitigation actions at the farm level.

The original dataset of 63 dairy farms (was ordered concerning the total environmental impacts (weighted results expressed in Pt) and divided into four groups corresponding to quartiles. One dairy farm has been selected and considered representative of each quartile only if its total weighted environmental impact was as close as possible to the quartile’s median.

Additionally, the study only considers five mitigation actions, and there may be other potential actions that could be effective in reducing environmental impacts. Despite these limitations, the study provides valuable insights into the potential mitigation actions for reducing the environmental impacts of the dairy industry.

The drafting of the manuscript involved a careful analysis of the most recent bibliography on the subject of actions to mitigate the environmental impact of cow's milk production. In this regard, the most easily applicable mitigation actions with environmental benefits at the farm level were identified. For these reasons, five mitigation groups containing various mitigation actions were considered, as listed in Table 1 of the manuscript.

The findings of the study could be useful for policymakers, dairy farmers, and other stakeholders in developing strategies to improve the sustainability of the dairy industry.

Overall, this is a well-conducted study that addresses an important environmental issue and provides practical solutions for reducing the environmental impacts of milk production.

Comments on the Quality of English Language

its just minor grammer issues 

Reviewer 2 Report

Dear authors, your article presents an adequate drafting and order of presentation of ideas. On the other hand, the subject matter is appropriate given the environmental context and the impact of all human activity on the sustainability of the world. I congratulate that these issues are approached from another perspective. After the review I would like to make the following comments.

In the introduction, lines 77 to 79, I recommend mentioning some of the few articles that have focused on mitigation strategies to enhance the importance of your writing.

Within the two objectives, in the first, why was this production cycle selected? Is it because of its importance for the country? It would be important to clarify it in the text.

The section on materials and methodology is adequately explained and does not present comprehension problems.

In the section of analysis of results and discussion I found it a little confusing to have a table of the previous section in the results section, I would recommend a rearrangement of the presentation of information to facilitate the reader's own reading since having a significant number of abbreviations throughout the text could probably improve comprehension.

The article fulfills both objectives throughout the discussion.

The conclusions are relevant, few works address from multiple aspects how to mitigate the impact of a manufacturing process of a product, as in this case the elaboration of cheeses with denomination of origin. I congratulate the authors and encourage them to generate more study models that contribute to the sustainability of the planet.

Best regards.

Author Response

Dear authors, your article presents an adequate drafting and order of presentation of ideas. On the other hand, the subject matter is appropriate given the environmental context and the impact of all human activity on the sustainability of the world. I congratulate that these issues are approached from another perspective. After the review I would like to make the following comments.

We thank you for your suggestions for improvements to manuscript foods-2362042. The manuscript has been edited in several parts.

In the introduction, lines 77 to 79, I recommend mentioning some of the few articles that have focused on mitigation strategies to enhance the importance of your writing.

Some references have been added, thank you for the suggestion. Please see line 79.

Within the two objectives, in the first, why was this production cycle selected? Is it because of its importance for the country? It would be important to clarify it in the text.

A few sentences have been added to the manuscript to underline northern Italy’s importance in producing PDO cheeses. Thank you for the suggestion. Please see lines 84-87.

The section on materials and methodology is adequately explained and does not present comprehension problems.

In the section of analysis of results and discussion I found it a little confusing to have a table of the previous section in the results section, I would recommend a rearrangement of the presentation of information to facilitate the reader's own reading since having a significant number of abbreviations throughout the text could probably improve comprehension.

Table 1 has been moved to section 2.3 Mitigation action selection. This move provides for a greater understanding of the table’s contents as suggested. Some rearrangements in the text have been implemented.

The article fulfills both objectives throughout the discussion.

The conclusions are relevant, few works address from multiple aspects how to mitigate the impact of a manufacturing process of a product, as in this case the elaboration of cheeses with denomination of origin. I congratulate the authors and encourage them to generate more study models that contribute to the sustainability of the planet.

Best regards.

Reviewer 3 Report

Specific comments and questions: 

1- In their analysis and discussion, the authors mention (lines214-217) that energy, bedding, and water used on the farm together account for 7% of the total impact. The most affected impact categories were CC (32%), WRD (25%), TE (11%), LU (7%), PM (6%), and POF (6%) (Figure 1), partially similar to Lovarelli et al., 2022. Is this a conclusion obtained from the literature reading? Or is it a conclusion reached by the authors through calculations?

2- Some important conclusions drawn in the analysis and discussion section are not mentioned in the abstract, and the authors are requested to revise the abstract section carefully.

3- Figure 2. Potential impact reduction by applying five mitigation actions in different scenarios for each quartile Q1, Q2, Q3 and Q4. Can they be redesigned? The important information cannot be read!

4- Table 1. How were the mitigation action groups considered, and the potential percent reduction of emitted substances calculated?

5- Where were the critical data obtained from the rest of the tables, Tables 2-3? In the second part of the paper, the specific research methods are not mentioned either!

Author Response

We thank you for your suggestions for improvements to manuscript foods-2362042. The manuscript has been edited in several parts.

1- In their analysis and discussion, the authors mention (lines: 214-217) that energy, bedding, and water used on the farm together account for 7% of the total impact. The most affected impact categories were CC (32%), WRD (25%), TE (11%), LU (7%), PM (6%), and POF (6%) (Figure 1), partially similar to Lovarelli et al., 2022. Is this a conclusion obtained from the literature reading? Or is it a conclusion reached by the authors through calculations?

1- The results are reported in paragraph 3.2. Original dataset and four quartiles of environmental impacts refer to the research conducted and compared with the scientific literature. Please see line 226.

2- Some important conclusions drawn in the analysis and discussion section are not mentioned in the abstract, and the authors are requested to revise the abstract section carefully.

2- Thank you for the suggestion, the abstract has been revised with additions.

3- Figure 2. Potential impact reduction by applying five mitigation actions in different scenarios for each quartile Q1, Q2, Q3 and Q4. Can they be redesigned? The important information cannot be read!

3- Information that was not directly available/understandable in Figure 2 was added to a new table in the supplementary materials (Table S1). The design of Figure 2 has the advantage of providing an overall overview of the results. Therefore, if possible, we would prefer to maintain the current figure structure, thank you.

4- Table 1. How were the mitigation action groups considered, and the potential percent reduction of emitted substances calculated?

4- The title of Table 1 has been changed for a better understanding of the content and to emphasize that it is derived from literature research.

5- Where were the critical data obtained from the rest of the tables, Tables 2-3? In the second part of the paper, the specific research methods are not mentioned either!

5- The specific research method mentioned in the Material and methods sections 2.1 and 2.2 have been briefly reported in the Result and discussion sections 3.1 and 3.2 to clarify the methods used.  Please see page 7, lines 196-197 and page 10, lines 216.
More specifically, Table 2 refers to the LCI with data obtained through surveys made at the farm level. Table 3 reports the LCIA characterized results performed following the PEF method and PEFCR rules for the dairy sector.